# Design and Characterisation of an Optical Fibre Dosimeter Based on Silica Optical Fibre and Scintillation Crystal

**DOI:** 10.3390/s22197312

**Published:** 2022-09-27

**Authors:** Michal Jelinek, Ondrej Cip, Josef Lazar, Bretislav Mikel

**Affiliations:** 1The Faculty of Electrical Engineering and Communication, Brno University of Technology, Technická 3058/10, 616 00 Brno, Czech Republic; 2Institute of Scientific Instruments of the CAS v. v. i., 612 64 Brno, Czech Republic

**Keywords:** gamma radiation measurement, silica optical fibres, dosimetry, fibre sensors, ionising radiation detection

## Abstract

In nuclear power plants, particle accelerators, and other nuclear facilities, measuring the level of ionising gamma radiation is critical for the safety and management of the operation and the environment’s protection. However, in many cases, it is impossible to monitor ionising radiation directly at the required location continuously. This is typically either due to the lack of space to accommodate the entire dosimeter or in environments with high ionising radiation activity, electromagnetic radiation, and temperature, which significantly shorten electronics’ lifetime. To allow for radiation measurement in such scenarios, we designed a fibre optic dosimeter that introduces an optical fibre link to deliver the scintillation radiation between the ionising radiation sensor and the detectors. The sensors can thus be placed in space-constrained and electronically hostile locations. We used silica optical fibres that withstand high radiation doses, high temperatures, and electromagnetic interference. We use a single photon counter and a photomultiplier to detect the transmitted scintillation radiation. We have shown that selected optical fibres, combined with different scintillation materials, are suitable for measuring gamma radiation levels in hundreds of kBq. We present the architecture of the dosimeter and its experimental characterisation with several combinations of optical fibres, detectors, and scintillation crystals.

## 1. Introduction

Currently, electronic or film dosimeters are used to measure ionising gamma radiation. Film dosimeters are not suitable for monitoring gamma radiation activity but only for measuring the total radiation dose. In contrast, electronic dosimeters allow continuous monitoring of gamma radiation activity and provide up-to-date information on the total radiation dose. The electronic dosimeter part with a photomultiplier or semiconductor detector is usually directly coupled to a suitable scintillation material or a semiconductor to detect ionising radiation [1,2]. High doses of ionising radiation have a direct destructive effect on any radiation-unshielded electronics. In particle accelerators, tokamaks, hot cells, or nuclear waste repositories, high levels of electromagnetic radiation, high temperature, and lack of space add up to these problems. Currently, these inaccessible and hazardous locations are addressed by indirect film measurements or electronic measurements at areas that are not hazardous to electronic systems and then calculating the activity at the desired location based on a model of the facility. No suitable measurement systems for these applications allow continuous measurement at such sites.

One way to continuously measure ionising radiation is to use a scintillation material that converts the incident ionising radiation into the visible part of the frequency spectrum and then transports this light to the electronic detector via optical fibres. These are available in silica and plastic materials. In recent years, their use in detecting various physical processes, i.e., in sensor technology, has increased significantly [3,4,5,6].

However, the use of optical fibres for the measurement of incident radiation, especially gamma radiation, is still marginal [7,8]. When optic fibres are exposed to ionising radiation, structural damage occurs inside the fibre, leading to additional attenuation of the optical signal. In silica fibres, the damage is mainly caused by the reaction of O–H or GE dopant bonds in the optical fibre and ionising radiation [9,10]. In plastic optical fibres, it is usually the interaction of one of the materials or compounds in the fibre with ionising radiation [7,11]. Plastic optical fibres often vary in material depending on the application and manufacturer. The commercial range of these fibres is extensive in terms of material composition and, in particular, the availability of plastic optical fibres with diameters ranging from hundreds of µm to tens of mm. Their principal disadvantage is higher attenuation than silica fibres [10,12,13,14].

For example, radiotherapy uses plastic optical fibres for radiation dose sensing. The fibres have a fundamental attenuation of tenths of dB/m, but when exposed to a moderate dose of ionising radiation, the attenuation gradually increases to tens of dB/m. Generally, most plastic optical fibres have a higher sensitivity to ionising radiation. However, new types are progressively emerging with increased resistance to the ionising radiation approach of silica optical fibres [15].

For optical fibres to measure ionising radiation continuously and over large distances from the measurement site, the base attenuation and attenuation after exposure to a radiation dose need to be as small as possible. Silica optical fibres are much more suitable for this purpose [15]. They exhibit lower base attenuation, and the worst results in their sensitivity to ionising radiation are still better or comparable to plastic optical fibres [16].

Nevertheless, in the fibre optic dosimeters published so far, the use of plastic optical fibres predominates [17]. Plastic fibres with larger diameters, typically 1–3 mm, are often used due to higher NAs, and they are not as fragile as silica optical fibres [18,19]. A common feature of these dosimeters is the adaptation of the diameter of the scintillation material to the diameter of the optical fibre [20,21], the deposition of scintillation material on the surface or inside the optical fibre [22], or the direct doping of the optical fibre with scintillation material [23,24]. Silica optical fibres, despite their advantages, are significantly less used in dosimeters, mainly due to their small diameters [25]. These dosimeters use the same principles as plastic optical fibres in the fibre–scintillation material connection. In addition, for very high levels of ionising radiation, simple monitoring of increasing optical fibre attenuation is also used [26]. In this paper, we present our fibre optic dosimeter that uses silica optical fibres to deliver the scintillation radiation from the ionising radiation sensor to the detectors. The architecture of the dosimeter incorporates the scintillation crystal sensors, interconnected with the optical fibre link to the photomultiplier or single photon counter we used to detect the scintillation radiation. The optical fibre link was constructed from commercial optical fibres, which needed special preparation before being fitted with SMA connectors and embedded into the stainless-steel armour. Compared with previously published dosimeters, we present a modular architecture where the scintillation material is not fixed to the optical fibre, and all parts of the dosimeter can be easily replaced based on the measurement requirements.

We have experimentally characterised the sensitivity of several silica optical fibres in combination with several scintillation materials. To compare these measurements, we developed sensor housings where the scintillation material can be interchanged, and the optical fibre can be attached so that the end of the fibre and the front of the scintillation crystal are always in the same relative position.

## 2. Materials and Methods (Methods and Instrumentations)

The design of the fibre optic dosimeter, shown in Figure 1, consists of three main parts: the fibre optic link, the ionising radiation sensor, and the detector. The main contribution of the dosimeter is the use of pure silica core optical fibres. Gamma radiation is converted into scintillation radiation with a wavelength in the visible part of the frequency spectrum in the scintillation crystal sensor. The exact wavelength depends on the type of scintillation crystal. The scintillation radiation is coupled into an optical fibre and delivered to one of the detectors, either a single photon counter (SPC) or a photomultiplier tube (PMT), which converts the radiation into electrical pulses. These pulses are then recorded by a frequency counter or sampled by an analogue-to-digital converter and recorded using a PC.

In this work, we investigated the proposed fibre optic dosimeter system with several combinations of components. The selection of components is summarised in Table 1, while detailed characteristics are presented later in respective sections.

### 2.1. Fibre Optic Link

Developing the optical fibre link included selecting suitable optical fibres and suitable optical connectors and integrating the protection of optical fibres against mechanical damage and permeating light from the outside environment.

The optical fibres were selected based on the following criteria: material (plastic vs. silica optical fibres), sensitivity to gamma radiation, mechanical and temperature resistance, core diameter, spectral range, and numerical aperture (NA). The first decision point was to choose between plastic and silica optical fibres. Plastic fibres are preferable in terms of better mechanical resistance and availability of larger diameters, but silica optical fibres are preferable for other parameters. Therefore, we limited our subsequent selection based on other parameters to silica optical fibres, considering that mechanical resistance can be increased by appropriate additional protection of the optical fibre and that the disadvantage of smaller available diameters is compensated by lower attenuation, higher radiation resistance, and resistance to higher temperatures in silica optical fibres [25].

The consequent selection was optimised for the maximum amount of scintillation radiation, coupled, transmitted, and detected. Probably the most significant issue with coupling the scintillation radiation into the optical fibre is that the single photons of scintillation radiation produced in the scintillation crystal due to incident gamma radiation have a random propagation direction. Therefore, it is impossible to concentrate the scintillation radiation into the optical focus of the optical fibre more efficiently by using additional optical elements. Based on our experiments, the most straightforward and cost-effective method is to adjoin the scintillation crystal in the sensor close to the optical fibre. Therefore, the optical fibre used should be as large in diameter as possible, ideally matching the diameter of the scintillation crystal and as high NA as possible.

We selected commercially available step-index multimode optical fibres, shown in Table 2, with pure silica cores of 0.4 mm, 1 mm, and 1.5 mm diameters and NAs of 0.37 and 0.5. The silica-core optical fibres FP1500URT, FP1000URT, and FP400URT have NAs of 0.5 and differ in their diameters, 1.5 mm versus 1 mm and 0.4 mm. The fourth optical fibre, F–MBE, has a smaller NA of 0.37. The FP1500URT, FP1000URT, and FP400URT optical fibres have a hard polymer cladding with an operating temperature range of −40 °C to 150 °C. The F–MBE optical fibre has a Hard class silicon cladding.

The next significant issue was to select suitable fibre optic link termination. The connectors should be mechanically robust and guarantee a constant distance between the optical fibre face, the scintillation crystal, and the detector chip. They should also be available for different fibre optic diameters. Based on these criteria, we selected SMA905 optical connectors with ferrule diameters matching the optical fibres. One of the reasons for this is the possibility of connecting optical fibres in laboratory conditions. The connectors ensure a sufficiently tight connection between the scintillation crystal and the detectors. This provided sufficient reproducibility of the individual measurements.

The bare fibres are first embedded in a shrink film of cross polyolefin, which protects the fibre surface to prevent ambient light penetration. Subsequently, the entire fibre is enclosed in stainless-steel armour that shields the optical fibre from mechanical stress. Both these protection measures contribute to the overall resilience of the optical fibre link, which is necessary in industrial-linked facilities and explosion-hazardous environments. Finally, the optical fibres with both protections applied are glued into the SMA905 connector with a two-part epoxy adhesive and then crimped, ground, and polished. For polishing, the bonded and refracted optical fibre in the connector is placed in a polishing wheel and polished on sandpaper with a roughness of 30 µm to 0.02 µm. Finally, the optical fibre in the connector is polished on polishing paper. The gradually improving surface quality at the optical fibre ends during the process is shown in Figure 2. Any unevenness or grooving at the end of the optical fibre reduces the amount of scintillation radiation coupled and, consequently, the amount of scintillation radiation delivered to the detection unit.

### 2.2. Ionising Radiation Sensor

Many scintillation materials are suitable for gamma-ray measurements, ranging from plastic, powder, organic, inorganic, and liquid to crystal. When using an optical fibre between the scintillation material and the detector, inevitably, only a portion of the generated scintillation radiation from the entire volume of the scintillation material can be coupled to the optical fibre. This fraction depends on the diameter and NA of the optical fibre. Thus, we focused on high-density crystal materials that can absorb a more significant fraction of the incident ionising radiation in a small volume. We selected the following scintillation materials: lanthanum bromide (LaBr3(Ce)), lutetium orthosilicate (LYSO(Ce)), thallium-doped sodium iodide (NaI(Tl)), and calcium fluoride (CaF2). Table 3 summarises the basic parameters of the crystals, including the thickness of the scintillation material needed for a 50% probability of conversion of gamma rays to visible scintillation radiation.

The following sensor design has already been performed for specific scintillation crystals and the connection to the fibre optical link with the SMA905 connector. The construction of the sensors consists of the scintillation crystal, a housing, and a fibre optic connection. The aluminium housing tightly encloses the scintillation crystal, preventing mechanical damage and irradiation by external light. The housing is designed so that there is an equal amount of construction material between the ionising radiation source and the scintillation material in all directions. The entire sensor consists of a sensor body with a scintillation crystal and a lid with an SMA905 connector for connecting the optical fibre. The fibre is threaded onto the sensor lid after the lid is closed (see Figure 3). This ensures that the end of the optical fibre and the front side of the sensor crystal are always in the same relative position. In addition, the scintillation crystal is covered on five sides with Teflon tape, which reflects more than 95% of the light in the light spectrum around 400 nm [27]. Due to the crystal’s small size, we could couple slightly more scintillation radiation into the optical fibre with this approach.

The ionising radiation sensor design uses solid, powder, or liquid scintillation materials. Each housing is easily customisable to adapt materials of different sizes and properties. The possible use of liquid scintillation materials offers broader but more technologically demanding possibilities for future coupling of optical fibre and scintillation material. This area is subject to further investigation.

### 2.3. Detectors

We used two transmitted scintillation detectors to develop and characterise the fibre optic dosimeter: the single-photon counter (LaserComponents) with a detection wavelength range from 380 nm to 500 nm and quantum efficiency of 45%. The output of the SPC is a preprocessed transistor logic signal (TTL) with a dark count rate of 10 counts/s. The SPC has an active area diameter of 100 µm with a typical dead time of 45 ns. The SPC has the advantage of small size and a 24 V power supply. This leads to more straightforward electrical wiring where a high voltage divider and other electronics are not required. The disadvantage is the smaller diameter of the active area and the associated complexity of concentrating the scintillation light from the optical fibre onto the active chip.

As a second detector, we used a photomultiplier tube (9266B, Et–Enterprises, Sweetwater, TX, USA) with a detection wavelength range from 290 nm to 630 nm and a quantum efficiency of 20%. It has a larger active area with an input window diameter of 51 mm compared with the SPC. The output signal is not preprocessed, so it potentially allows for performing a spectral resolution of the measured scintillation radiation [28,29]. In contrast, it requires a high–voltage source and associated electronic subsystems.

Both detectors were equipped with a stainless-steel spacer on the input side and a thread for the SMA905 connector, which guaranteed a constant distance of the optical fibre end from the active surface of the detectors.

## 3. Measurements and Results

For the measurements during the period of the design and subsequent characterisation of the optical fibre dosimeter, we used ^60^Co and ^137^Cs gamma radiation sources (Eurostandard-cz). A list of nuclides with different activities is given in Table 4. The activity of the ionising radiation sources decreases with time, following $N/N_{0}=exp\left(-\lambda t\right)$, where *N* is the number of remaining atoms, *N_0_* is the number of atoms at time *t* = 0, and λ is the radioactive decay constant. In the presented data, for each measurement, the source’s nominal activity is always interpolated from the reference value to the value valid for the day of measurement.

### 3.1. Experimental Procedure and Pilot Tests

We characterised and optimised the optical fibre link during the developing process and compared the dosimeter measurement sensitivity achieved with the two different scintillation detectors. We used the 1-meter-long optical fibre link for these tests with the FP1500URT optical fibre, which has the largest diameter and the largest NA. We used an LYSO scintillation crystal sensor to detect ionising radiation. It does not have the highest light yield of the crystals used, but due to its highest density, we assumed the best efficiency in converting ionising radiation to scintillation.

In the first test, we sequentially connected the optical fibre from the sensor to the SPC and PMT detectors. We set the detection threshold level for the SPC detector to 0.5 V and the gate time to 1 s. We set the supply voltage for the PMT detector to 1750 V and the detection threshold level to 20 mV. For each measurement, we chose an irradiation time of 600 s. Using a shorter time, potential errors in the conversion of light to electrical pulses or sudden changes in the power level could significantly affect the measurement. On the other hand, a long time does not increase the stability of the measurement. Therefore, in the following measurements, we calculated the arithmetic average for the detected scintillation photons over the 600 s measurement period to obtain the system response (sensitivity) to irradiation with the particular sensor, optical fibre link, and detector. The sensitivity is expressed as the source activity’s counts per second (countrate).

The raw data of the real-time output signals of both detectors during the measurement of the ionising source ID1, shown in Figure 4, revealed a stable SPC countrate with an average of 716.5 cps and a standard deviation of 29.02 cps; the PMT revealed a stable countrate with an average of 1939.5 with a standard deviation of 49.68 cps.

With the aforementioned experimental procedure, we further investigated the sensitivity using both detectors and several gamma radiation sources. The results in Figure 5 indicate that both detectors’ sensitivity is roughly comparable. The efficiencies of the PMT and SPC are 7.73 and 2.89 cps/kBq, respectively. Due to the larger detector area diameter, the photon countrate from the PMT is 2.5 times higher, although the quantum efficiency of 20% is lower than the SPC. Thus, the PMT comes out as a more sensitive detector and allows a more detailed analysis of the detected photons due to the direct output without additional electronic elements.

### 3.2. Characterisation of the Connector Face Preparation

In the design of the fibre optic dosimeter, much of our attention was paid to creating the fibre optic link. In the final preparation of this splice, we were involved in grinding and polishing the connector ferrule faces to increase the amount of scintillation radiation coupled. To verify this theory, we performed successive measurements of the transmitted scintillation radiation after each grinding and polishing step (shown in Figure 2) for FP1500URT and FP1000URT optical fibres. For the measurements, we used an SPC detector, an LYSO scintillation crystal, and an ID1 ionising radiation source, see Figure 6. The dependence of the dosimeter measurement sensitivity on the roughness of the fibre optic faces was measured after each polishing step during fibre link termination. In these steps, the transmission properties of the optical fibre are gradually improved. After polishing, the fibre surface is glossy, with minimal visible nicks and scratches that would impair scintillation light coupling.

### 3.3. Characterisation of the Optical Fibre Dosimeter with Different Scintillation Crystals and Optical Fibre Links

We investigated several combinations of different sensors and different optical fibre links. We used the SPC detector with lower sensitivity, see Figure 7. This detector’s lower sensitivity allowed us to find the dosimeter measurement sensitivity limits, which can then be extrapolated with the PMT detector. As in the previous test, the observed sensitivity is the arithmetic average of the scintillation photons detected per 600 s, with the detection threshold set to 0.5 V and a gate time of 1 s. The dosimeter was permanently connected according to the principle in Figure 1. We used gamma ionising radiation sources of ^60^Co and ^137^Cs with activities according to Table 4, which were recalculated on the measurement date of each measurement. For the sensors, we used scintillation materials LYSO(Ce), LaBr3(Ce), NaI(Tl), and CaF2(Eu). The scintillation radiation from the crystals was delivered through fibre optical links with silica optical fibres FP1500URT, FP1000URT, FP400URT, and F–MBE with a range of diameters, lengths, and NAs.

The difference in sensitivity among different optical fibres (all of them 1-m-long) with the LYSO crystal sensor and ID1–ID3 ionising radiation sources is shown in Figure 7. The FP1500URT optical fibre is the most efficient in coupled, transferred, and detected scintillation radiation from the sensor among the compared optical fibres. The efficiency is 2.89 cps/kBq compared with other fibres with efficiencies of 1.37, 0.61, and 0.18 cps/kBq. This is due to its largest NA and largest core diameter. This advantage is partially offset by the disadvantage of a smaller bending radius, which may limit its use in some industrial applications. Other fibres with smaller diameters can also be bent in smaller diameters but can couple and transmit a smaller fraction of the scintillation radiation. These advantages and limitations have to be considered in every application. Optical fibres with the same core diameter, F–MBE and FP1000URT, have different amounts of coupled, transmitted, and emitted scintillation radiation due to different NAs.

Figure 8 compares the sensitivity for different scintillation crystals: LYSO(Ce), LaBr3(Ce), NaI(Tl), and CaF2(Eu) irradiated with reference sources ^60^Co ID1–ID3 (Figure 8a) and ^137^Cs ID4–ID6 (Figure 8b). We used FP1500URT optical fibre with a length of 1 m because it delivers the most scintillation radiation. The results indicate that an LYSO is the most suitable scintillation material in the gamma radiation conversion efficiency for detecting ionising radiation of ^60^Co and ^137^Cs due to its light yield and density combination. 

The efficiency difference between the best scintillation crystal and the worst, i.e., LYSO vs. NaI(Tl), is 2.89 vs. 0.52 cps/kBq for ^60^Co and 1.00 vs. 0.18 cps/kBq for ^137^Cs, respectively.

Figure 9 compares the FP1500URT 1 and 5 m optical fibre measurements. We used the ID1–ID3 ionising radiation sources for the measurements and the LYSO scintillation crystal sensor. The optical fibre length of 5 m was chosen because it is already sufficient for separating sensors and electronics in most potential applications safely. The comparison shows that a 5 m optical fibre has a sensitivity of 1.64 cps/kBq due to its higher attenuation, while a 1 m optical fibre has a sensitivity of 2.89 cps/kBq.

All measurements together are presented in Figure 10. It shows the different conversion efficiencies of ionising gamma radiation in different scintillation materials in conjunction with different types of optical fibres and their different lengths. It is clear that from the point of view of the dosimeter measurement sensitivity, the most sensitive is a combination of the FP1500URT optical fibre link and the sensor with an LYSO scintillation crystal. On the contrary, the least sensitive measurement is the configuration with FP400URT, which has a smaller diameter when coupled with sensors with NA(Tl) and CaF2(Eu) scintillation crystals.

Nevertheless, our measurements have shown that the designed optical fibre dosimeter can measure ionising gamma radiation with activity from tens of kBq with all these silica optical fibres, detectors, and scintillation crystals. Practically, the dosimeter can be adapted to most possible applications by a suitable selection of individual components and their combinations.

Compared with the fibre optic dosimeters presented so far, the SPC detector configurations of the presented dosimeter are approximately five-times less sensitive. On the contrary, using PMT, the sensitivity of the presented dosimeter is comparably sensitive or more sensitive [19,20]. Thus, the differences in sensitivity are mainly due to the sensitivity of different detectors and the different optical fibre properties, diameters, and NAs.

## 4. Conclusions

We have introduced a fibre optic dosimeter designed to measure ionising gamma radiation in environments with high radiation levels, radio frequency and electromagnetic interference, and space constraints. The optical fibre dosimeter consists of a gamma radiation sensor with a scintillation crystal, a silica multimode optical fibre link, a scintillation radiation detector, and a control PC. We developed a methodology to fabricate the optical fibre link using different fibre types, which includes termination with SMA connectors and a mechanical embedding into an armoured jacket that increases mechanical resilience and prevents ambient light penetration.

We tested the dosimeter measurement sensitivity using different combinations of components: four different silica multimode optical fibres ranging from 0.4 mm to 1.5 mm in diameter with NAs of 0.37 and 0.5; four gamma-ray sensors with scintillation crystals (LYSO(Ce), LaBr3(Ce), CaF2(Eu), and NaI(Tl)); two different scintillation radiation detectors—a single photon counter and a photomultiplier tube.

We have demonstrated the feasibility of using the dosimeter with all prepared optical fibres, sensors, and detectors. The best dosimeter measurement sensitivity was achieved using a 1.5 mm diameter optical fibre with an NA of 0.5, a sensor with an LYSO crystal, and a photomultiplier. In particular, the photomultiplier has the advantage of more sensitive scintillation detection, but the single photon counter is only slightly inferior and easier to use for most applications. Measurements have also shown that silica optical fibres with smaller optical core diameters and NAs can be used to measure higher ionising radiation activities. We also demonstrated good compatibility of different silica optical fibres with all measured scintillation crystals. The developed optical fibre dosimeter can, thus, be easily used for gamma radiation measurements in a wide range of industrial and safety applications. With easily interchangeable sensors, optical fibres, and detectors, the system can be adapted to measure the dose rates over a wide range in various critical environments.

## Figures and Tables

**Figure 1 sensors-22-07312-f001:**
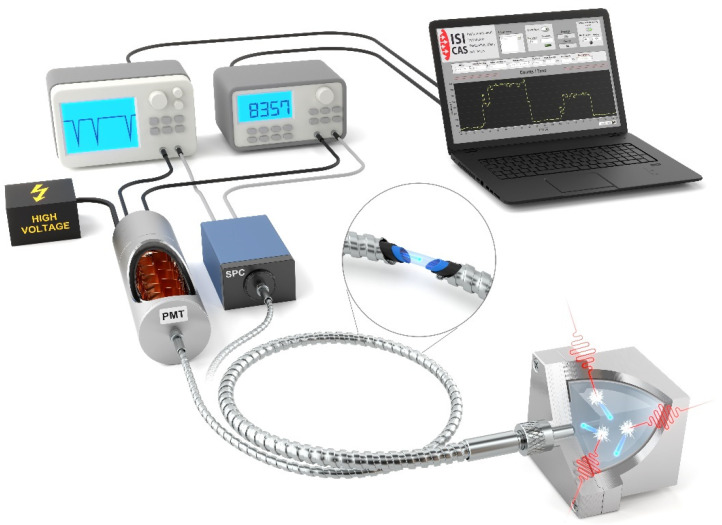
Schematic of the measuring system for ionising radiation measurement. The measuring system contains two variable detection units—SPC and PMT, multimode optical fibre and scintillation crystal hidden in an aluminium box.

**Figure 2 sensors-22-07312-f002:**
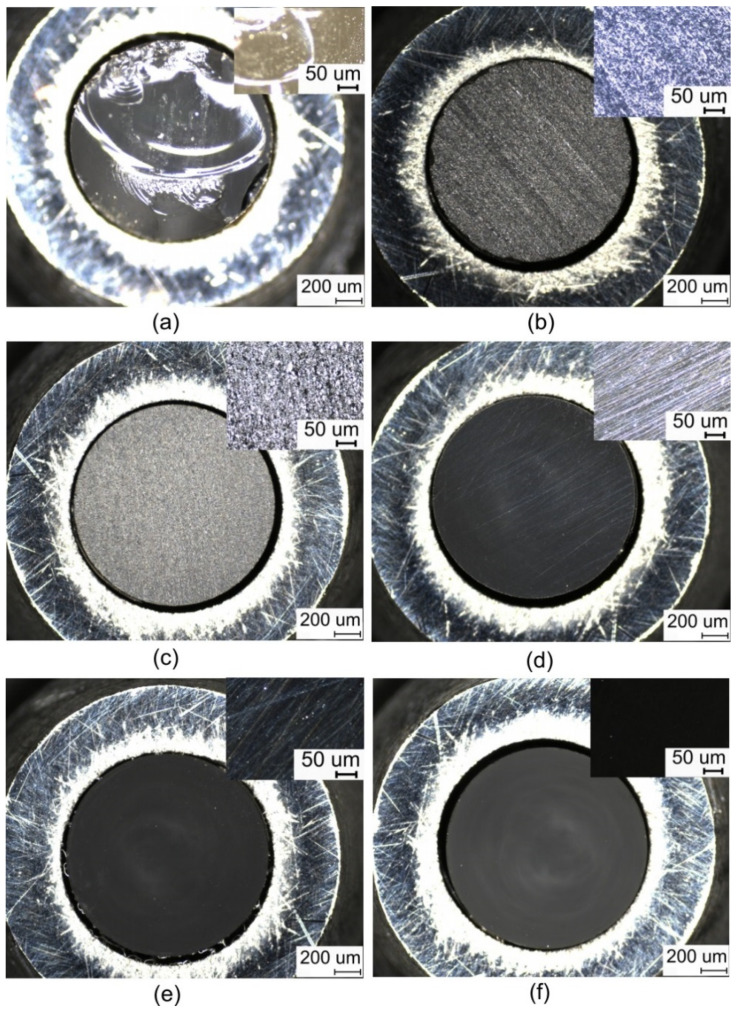
The process of grinding/polishing optical fibres with a detailed section, (**a**) cleaved optical fibre, (**b**) polished with 30 µ abrasive paper, (**c**) polished with 6 µ abrasive paper, (**d**) polished with 3 µ abrasive paper, (**e**) polished with 1 µ abrasive paper, (**f**) polished with fine abrasive paper.

**Figure 3 sensors-22-07312-f003:**
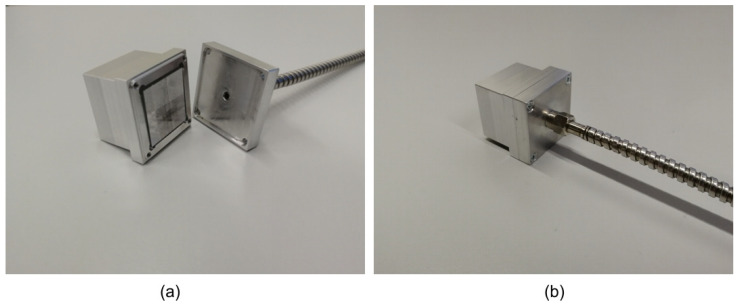
Design of the sensor: (**a**) open sensor construction with scintillation crystal; (**b**) closed sensor construction with connected optical fibre.

**Figure 4 sensors-22-07312-f004:**
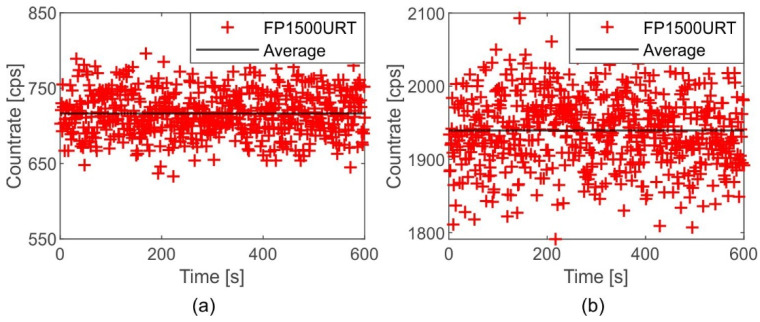
Sample raw data output signals from (**a**) the SPC and (**b**) the PMT detectors for 600 s. The graph also shows the average observed countrate.

**Figure 5 sensors-22-07312-f005:**
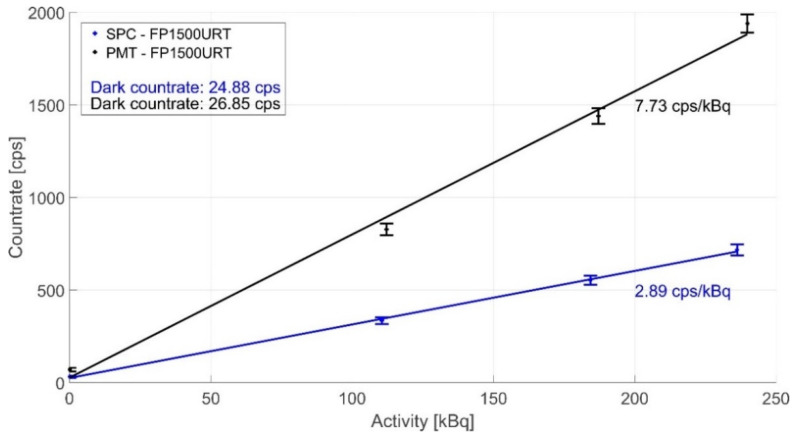
A comparison of the averaged data from SPC and PMT detectors in the optical fibre dosimeter configuration with an LYSO(Ce) scintillation crystal and FP1500URT optical fibre with a length of 1 m and ID1–ID3 gamma radiation source.

**Figure 6 sensors-22-07312-f006:**
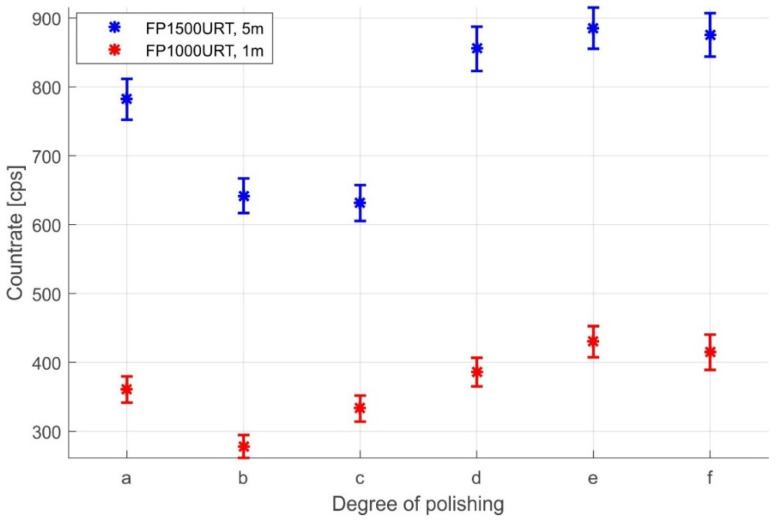
The amount of coupled, transferred, and emitted scintillation radiation depends on the roughness levels for the FP1500URT and FP1000URT. The SPC detector detected the scintillation radiation.

**Figure 7 sensors-22-07312-f007:**
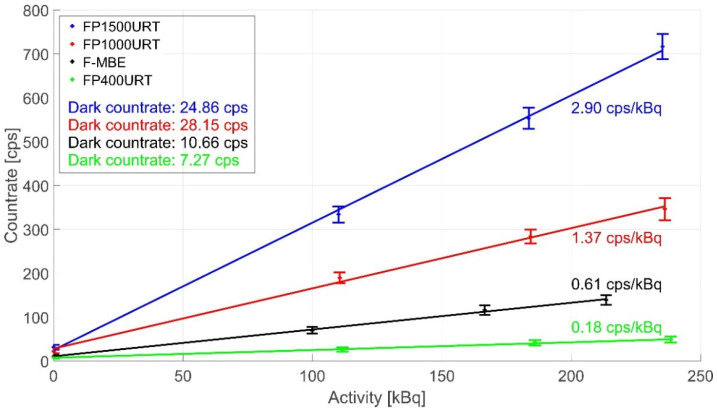
Dosimeter measurement sensitivity comparison using the LYSO(Ce) crystal and different optical fibre links of 1 m length under irradiation of ID1–ID3 sources.

**Figure 8 sensors-22-07312-f008:**
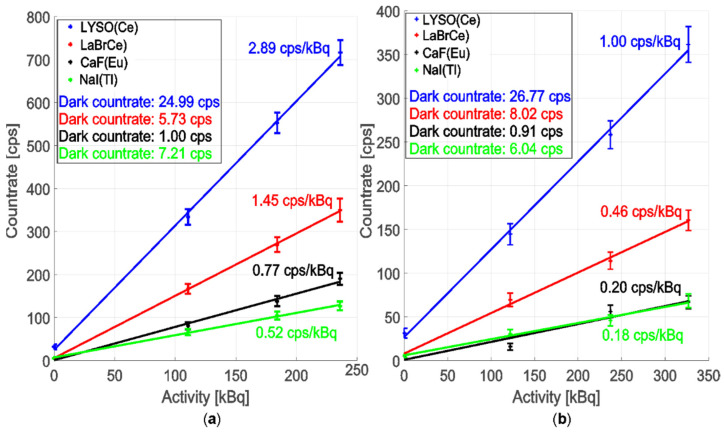
Dosimeter measurement sensitivity comparison with the scintillation crystal coupling to FP1500URT optical fibre with 1.5 mm optical core diameter and 1 m length. (**a**) The ionising radiation sources: ID1–ID3. (**b**) The ionising radiation sources: ID4–ID6.

**Figure 9 sensors-22-07312-f009:**
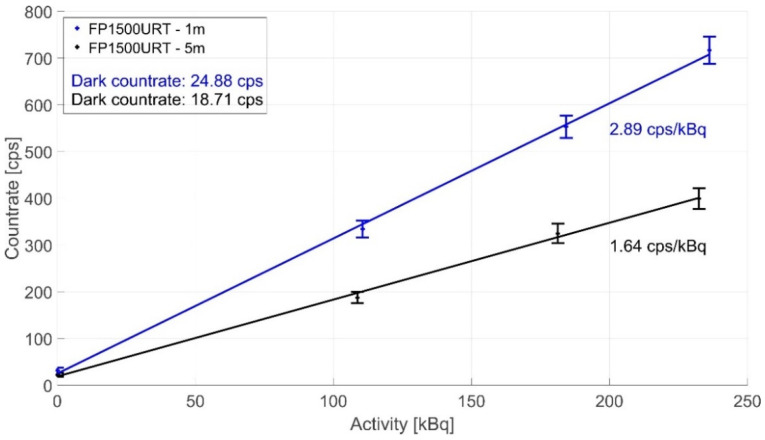
Dosimeter measurement sensitivity decreasing in scintillation radiation delivery with different lengths of the optical fibre link (FP1500URT silica optical fibre) during irradiation of ID1–ID3 sources.

**Figure 10 sensors-22-07312-f010:**
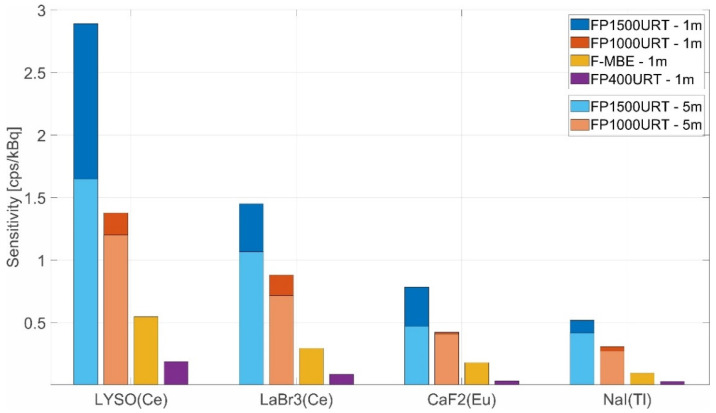
Different conversion efficiencies of ionising gamma rays—^60^Co in different scintillation materials in conjunction with other types of optical fibres and their different lengths.

**Table 1 sensors-22-07312-t001:** Combination of all measurement options.

Detection Unit	Scintillation Crystal	Optical Fibre Link	Optical Fibre Link Length	Ionising Source
↳SPC	↳LYSO(Ce)	↳FP1500URT	↳1 m	↳^60^Co
↳PMT	↳LaBr3(Ce)	↳FP1000URT	↳5 m	↳^137^Cs
	↳NaI(Tl)	↳FP400URT		
	↳CaF2(Eu)	↳F-MBE		

**Table 2 sensors-22-07312-t002:** Overview of the properties of the optical fibres used.

Fibre	Core Diameter (µm)	Cladding Diameter (µm)	NA	Core Material	Material Cladding	Operating Temperature (°C) <min; max>
FP1500URT	1500	1550	0.5	Pure silica	Hard Polymer	<−40; 150>
FP1000URT	1000	1035	0.5	Pure silica	Hard Polymer	<−40; 150>
FP400URT	400	425	0.5	Pure silica	Hard Polymer	<−40; 150>
F-MBE	1000	1035	0.37	Pure silica	Hard class Silica	<−65; 125>

**Table 3 sensors-22-07312-t003:** The scintillation materials used to measure ionising radiation.

Scintillator	LYSO(Ce)	LaBr3(Ce)	NaI(Tl)	CaF2(Eu)
Dimensions (mm)	20 × 20 × 20	16 × 16 × 20	Ø25.4 × 25.4	20 × 20 × 20
Light yield (photons/keV)	33	63	35	19
Thickness to stop 50% of 662 keV photons (cm)	1.1	1.8	2.5	2.9
1/e Decay time (ns)	36	16	250	940
The wavelength of max emission lm (nm)	420	380	415	435
Refractive index at lm	1.81	1.9	1.85	1.47
Density (g/cm^3^)	7.1	5.08	3.67	3.18

**Table 4 sensors-22-07312-t004:** The list and the parameters of ionising sources used for experimental characterisation.

ID	Radionuclide	Type	Reference Date (DD.MM.YYYY)	Reference Activity (kBq)	Uncertainty of Activity (kBq)
1	^60^Co	EG 3X	08. 07. 2019	318.0	1.6
2	^60^Co	EG 3	08. 07. 2019	248.1	1.7
3	^60^Co	EG 3	08. 07. 2019	148.8	0.7
4	^137^Cs	EG 3	08. 07. 2019	344.2	2.8
5	^137^Cs	EG 3X	08. 07. 2019	249.8	2.0
6	^137^Cs	EG 1X	08. 07. 2019	128.3	1.0

## Data Availability

Not applicable.

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
