# Peer review of "Design and Characterisation of an Optical Fibre Dosimeter Based on Silica Optical Fibre and Scintillation Crystal"

_sensors, 2022, doi:10.3390/s22197312_

Round 1

Reviewer 1 Report

The publication is very good, the reviewer recalls for printing. Of course, a reference could be added to the use of avalanche photodiodes to detect small signals, but this is not essential.

Author Response

Thank you for reviewing our manuscript. We agree with your comment; the reference to the use of avalanche photodiodes is a benefit of the article. We added reference 2 - Jiang, W.; Chalich, Y.; Deen, M.J. Sensors for Positron Emission Tomography Applications. Sensors 2019, 19, doi:10.3390/s19225019, where the avalanche photodiodes and other highly sensitive detectors are described for measuring low signals from measurement positrons.

Reviewer 2 Report

The authors of the paper present quite interesting results. In my opinion, they are of sufficient practical importance and will be of interest to a wide range of readers of the Sensors journal. Despite this, I would draw attention to a few minor flaws, the correction of which is highly desirable:

1. The authors describe a sufficient number of variations of their own installation, where they change the type of fiber, the sensitive element, and other parameters, but do not describe the comparison of the system sensitivity with the results obtained by other authors. In the introduction, in refs [6,7], such information is given. I believe that these conclusions should be presented in the final part of the article.

2. The experimental setup is shown in the figure at the beginning of the paper. At the same time, the description of the technical characteristics of its elements is distributed throughout the document. I suggest combining them into a separate table.

3. Some sections of the manuscript end with figures or tables, I recommend placing them in the text, Immediately after the first mention.

Author Response

Point 1: The authors describe a sufficient number of variations of their own installation, where they change the type of fiber, the sensitive element, and other parameters, but do not describe the comparison of the system sensitivity with the results obtained by other authors. In the introduction, in refs [6,7], such information is given. I believe that these conclusions should be presented in the final part of the article.

Response 1: Thank you for reviewing our manuscript. We added a partial comparison of the result with other authors to the final part of the article. The complete comparison is complicated due to using different detectors, optical fibres and various detection techniques. Many authors use other radiation sources and energies, the spectral resolution of the measured signals and various scintillation materials. We choose articles that are a comparable setup of measurement, optical fibres, scintillation material and detectors.

Point 2: The experimental setup is shown in the figure at the beginning of the paper. At the same time, the description of the technical characteristics of its elements is distributed throughout the document. I suggest combining them into a separate table.

Response 2: We agree with your comment. The separate table close to the main schema of the measurement adds clarity to the article. We have added a table that summarises and clearly shows the options for configurations of the measurement system.

Point 3: Some sections of the manuscript end with figures or tables, I recommend placing them in the text, Immediately after the first mention.

Response 3: Thank you for your comment. We have followed basic typographic guidelines where it is preferable to place images at the top or bottom of the page if possible. This caused the figures to move further away from the first mention, which reduced the article’s clarity. We have attempted to correct this by moving figures 4-10.

Author Response

Point 1: The concept of a scintillator-fibre-detector dosimeter is not new, and many papers have already been published on this topic in the literature. The present manuscript must be placed among the previous work done in this field. This important point is not currently addressed in the article introduction, which deals with film and electronic dosimeters, using optical fibers for other types of sensing, and the radiation effects on optical fibers.

Response 1: Thank you for reviewing our manuscript. We definitely agree with your comment. During the preparation and subsequent editing of the article, it was omitted of presenting the current state of fibre optic dosimetry. We have incorporated an extension to the introduction dedicated to this issue. We have added ten references, which refer to review publications and other current methods.

Point 2:

l.17 : “deliver the radiation”. To be more precise, I suggest to add “deliver the scintillation radiation”.

  1. 64 : silicon or silica ?
  2. 295 : “We used the SPC detector with the lower sensitivity, see Figure 3.” Please check if Figure 3 is the right one to be mentioned here.

Figure 8a : the vertical axis label is “Sensitivity [cps/kBq]”. Please check whether it may be “Countrate [cps]”, like on Figure 8b.

Response 2: Thank you for your attention to the article; we apologise for these errors that remained in the article despite all efforts. We have corrected all of them.
